# The Impact of School Closures on Service Utilization in School-Based Health Centers

**DOI:** 10.3390/ijerph20054588

**Published:** 2023-03-05

**Authors:** Eleanor Castine Richards, Madelyn R. Allen, Margaret Danielle Weiss

**Affiliations:** Cambridge Health Alliance, Harvard Medical School, Cambridge, MA 02139, USA

**Keywords:** COVID-19, mental health, youth, school mental health, pandemic, access, telehealth, adolescent

## Abstract

Background: The pandemic was followed by a severe mental health crisis in youth with both an increase in the prevalence of mental health problems and a decrease in requests for and access to care. Methods: data were extracted from the school-based health center records in three large public high schools that include under-resourced and immigrant communities. Data from 2018/2019 (pre-pandemic), 2020 during the pandemic, and then in 2021 after the return to in-person school were compared regarding the impact of in-person, telehealth, and hybrid care. Results: Despite the increase in mental health needs globally, there was a dramatic decrease in referrals, evaluations, and the total number of students seen for behavioral health care. The time course of this decrease in care was specifically associated with the transition to telehealth, although treatment did not return to pre-pandemic levels, even after in-person care became available. Conclusions: Despite ease of access and increased need, these data suggest that telehealth has unique limitations when delivered in school-based health centers.

## 1. Introduction

The COVID-19 pandemic resulted in a loss of peer connection, disruption to daily schedules, and health, academic, emotional, and economic stressors for youth. This led in turn to an increase in emotional and behavior problems of up to 43% [1,2,3]. Research indicates that the stressors brought about by COVID-19 selectively impacted adolescents more than toddlers or latency-aged children [4,5] and further selectively impacted under-resourced populations of youth, especially those with prior exposure to adversity [3,6]. At the present time, over half of adolescents need mental health care, and symptoms of depression and anxiety among youth are at an all-time high [7,8,9,10]. One report from Virginia schools found that mental health as the top health concern among students almost doubled from 15% to 27% during the pandemic, while available services decreased [11]. 

Despite the increased need resulting from the youth mental health crisis, the WHO (2020) estimates that the pandemic disrupted up to 72% of mental health services to youth, and a Medicare study in 2020 found a 44% decrease in mental health services, [12,13] cognitive development screenings, and a decline in emergency department visits [14], especially for Black youth [15]. When schools closed in March 2020, the main concern was the potential for a loss in academic skills. The impact of school closure on youth mental health and access to mental health services was not yet apparent [14]. Only now are we seeing the full impact of the ‘echo pandemic’ or the collateral damage of the pandemic on mental health. This paper explores the immediate impact of the pandemic on the demand for and provision of behavioral health services in high schools. 

More than half of all schools offer some type of mental health service, and more than a third of adolescents who are engaged in mental health services receive them exclusively through school programs and clinics [16,17]. These adolescents are disproportionately youth with lower family income, public health insurance, and from racial and ethnic minority backgrounds. School-based health centers (SBHCs) provide access to primary care, mental health, and other health services to over six million students in the United States [18]. Very little has been reported on how this changed during the pandemic. By April 2020, school was suspended in 188 countries, and over 90% of enrolled learners (ranging from one to five billion youth) were no longer receiving in-person education [19]. Subsequently, these youth no longer had access to in-person school services including SBHCs. This paper looks specifically at SBHC provision of mental health care [19].

SBHCs have been on the rise over the last two decades and greatly increase access to care for youth and thus improve physical and mental health outcomes broadly, as well as promote equity [16,18]. Studies suggest that youth are significantly more likely to complete mental health treatment that is provided in a school setting and more than half of youth who access mental health services generally are doing so by way of their school [17,20]. One study found that adolescents were 21 times more likely to initiate visits for mental health reasons at SBHCs compared with community-based facilities [21]. Further, youth of color have been found to utilize SBHC services more often than other community health centers [21]. SBHCs destigmatize obtaining help for mental health, eliminate transportation barriers, and provide on-site access and collateral in the setting where the youth spend most of their day [18]. 

When SBHCs shut down in-person services, youth who were dependent upon their services lost access to care [22]. In one survey of 427 SBHCs nationwide, 77% of SBHCs closed temporarily, 5% closed permanently, and only 12% remained physically open. Nevertheless, over half of the SBHCs (57%) initiated or increased telehealth services [22]. A recent qualitative study looked at changes to the provision of services and priorities for the 2021–2022 year and found increased acuity of mental health problems, greater immediacy of need, and greater complexity and comorbidity in the face of continued lack of staff [22]. 

The first study to look at changes in service utilization broadly, and mental health specifically, looked at a network of 180 SBHCs in Connecticut and found a 12.3% increase in behavioral vs. medical service utilization pre-pandemic vs. pandemic [23]. This study does not provide information on the absolute rates of mental health service utilization during the pandemic, which would be driven by both the decreased SBHC utilization and the relative increase in mental health vs, medical care. 

The difference in quality between in-person and telehealth delivery of school-based mental health care may be more meaningful than the difference between in-person, telehealth, or hybrid care in other types of outpatient mental health settings. Telehealth school-based mental health care compromises many of the unique advantages that contribute to the success of school-based mental health [24]. Telehealth delivery of school-based mental health is not compatible with ‘dropping in.’ Youth who might otherwise have wandered into the SBHC, may be much less likely to put in the effort to make an appointment. Teachers who no longer have the opportunity for informal chit chat with students, may be less aware of their distress and less likely to suggest they obtain help. Youth who did not participate in online learning may have dropped off the radar of teacher’s awareness. The emergence of the ‘pandemic missing’ children who never returned to school and are not registered as being home schooled or in private school is a further serious outcome of the impact of the disengagement from in-person attendance [25,26]. A recent analysis found approximately 230,000 students in 21 states, including Massachusetts, whose absences could not be accounted for and are being referred to as the ‘pandemic missing’ [27]. Closure of in-person school and in-person behavioral health care also selectively impacts students with special needs who receive specialized services that cannot effectively be delivered remotely such as ABA for autism or occupational therapy. 

If we believe that the most effective way to increase access to mental health care for youth, particularly those who are under-resourced, is through schools, then it is essential to document and analyze how the pandemic impacted both referrals for mental health service and provision of mental health service as an outcome in its own right. Our objective in this study is to analyze change in the provision of mental health services pre-pandemic vs. during the pandemic at SBHCs in three large, urban Massachusetts public high schools. These findings help to identify the role of school-based mental health care during the pandemic, and to evaluate the impact of telehealth vs. in-person behavioral health services within the SBHCs [28,29].

## 2. Methods

The data were drawn from the Cambridge Health Alliance (CHA) electronic medical records. Cambridge Health Alliance is a safety net, community-based hospital which serves three SBHCs in three urban communities adjacent to Boston: Cambridge, Everett, and Somerville. Seventy-four percent of patients served by Cambridge Health Alliance are insured with Medicaid. Table 1 includes demographic information of the student population at the three respective schools from 2020 to 2021, as reported by the Massachusetts Department of Education. 

Data were collected from 1 January 2018 to 31 December 2021, using Epic Slicer Dicer software. The patient population was defined as all youth who were seen in any of the three school-based health centers. Data extracted included: the total number of patients seen by year and by month for behavioral health services, the number of new referrals, the numbers of completed evaluations, terminations, and transfers. A referral for behavioral health care can be placed by those within the CHA health care system such as a primary care physician or another behavioral health provider (e.g., a psychiatrist refers a patient for therapy; the inpatient team refers a patient for outpatient therapy). A referral can also be made by adults outside of the system such as school staff, families, the department of children and families, and more. Youth may self-refer but parental consent is required for the patient to be scheduled for an appointment. Terminations may occur when treatment is completed, the patient elects to stop treatment, the patient drops out, or the patient is referred out for more specialized services. Transfers occur when a patient switches to a new provider or service within CHA. Given that schools closed in the middle of March 2020, the month of March 2020 was excluded since both pandemic status and modality of treatment are confounded between the middle and end of the month. 

The institutional review board waived approval because this was a retrospective review using de-identified aggregate data. 

## 3. Results

### 3.1. Patients Seen by Year

The number of patients seen in the three SBHCs was stable up until the start of the COVID-19 pandemic. All three school districts ceased in-person school the third week of March 2020, which coincided with the closure of all in-person physical and behavioral health care in the SBHCs. The clinics continued to receive referrals and intakes were conducted by telehealth (i.e., via phone or video) instead of in-person. The schools reopened full time in-person at varying points in the spring of 2021. At that time, the SBHCs opened for physical health care, but behavioral health services continued to be offered through telehealth. In the fall of 2021, the SBHCs switched to a hybrid model in which students were able to access both in-person and telehealth. 

In order to capture larger trends over time, we examined patient service by year, comparing patient numbers for 2018–2019 and 2020–2021. There was a 34% decline in the average number of patients seen between 2018 and 2019 and 2020–2021 (see Figure 1).

### 3.2. Patients Seen by Month

Since school closure does not align with the school year, the impact of the pandemic and modality of service was analyzed by month (see Figure 2). The switch from in-person care to telehealth for behavioral health services occurred in mid-March 2020 and continued until September 2021, following which both telehealth and in-person services were offered. Service modalities over time are defined as: 1. in-person only, 2. telehealth only, and 3. hybrid (i.e., any combination of in-person and telehealth services). Since March 2020 was a transition period, it was excluded from the analysis. 

We conducted an ANOVA with service modality (in-person only, telehealth only, hybrid) as the independent variable and found a statistically significant difference *F*(2, 44) = 25.82, *p* < 0.001. Gabriel post-hoc tests revealed that telehealth only (*M* = 50.35, *SD* = 27.79) resulted in a significantly lower number of patients being seen for behavioral health services compared with in-person only (*M* = 126.31, *SD* = 39.58) and hybrid (*M* = 123.75, *SD* = 21.30). There was no significant difference between in-person and hybrid. There was a 60% decrease in the average number of patients seen for behavioral health treatment when modalities of treatment changed from in-person care to telehealth. 

ANOVA analysis of pre-pandemic vs. pandemic status was also statistically significant *F*(1, 45) = 28.55, *p* < 0.001 with more patients being seen prior to the onset of the pandemic (*M* = 126.31, *SD* = 39.58) compared with during the pandemic (*M* = 64.33, *SD* = 39.47). 

### 3.3. Behavioral Health Referrals, Evaluations, Terminations, and Transfers

Behavioral health referrals and new evaluations declined steeply following the onset of the pandemic (see Figure 3). There was a 70% decrease in referrals with 254 referrals for school-based behavioral health services in 2018 and 76 in 2020. Even after the schools and SBHCs returned to in-person behavioral health care, referrals did not reach pre-pandemic levels. There were 160 referrals in 2021 compared with 254 referrals in 2018, demonstrating a 37% decrease overall. In 2021, our data demonstrated that the number of referrals outpaced the number of evaluations for the first time since 2018. At the same time, the number of terminations and transfers, both of which signify the end of care, decreased collectively, resulting in a waitlist for behavioral health care. 

### 3.4. Number of Patient Visits

Between 2019 and 2020, there was an absolute decrease in the number of patient visits. The highest number of visits (4855) occurred in 2019 with the lowest number of visits (2191) taking place in 2020. Even after the schools and SBHCs reopened, there was no substantial improvement in patient visits: there were 2601 visits in 2021. 

## 4. Discussion

Our findings are relevant to understanding the impact of the pandemic on youth mental health, the demand for service, and the relative effectiveness of in-person, telehealth, and hybrid modalities of treatment in SBHCs. Our data indicate that closure of in-person services had a deleterious impact on access to care despite the switch to telehealth. Our analysis adds to the literature in providing a fine-grained picture of the impact of the pandemic on mental health and the impact of telehealth which is not available in national data sets. 

We hypothesized that with the onset of COVID-19 and the increase in emotional distress among students, the number of referrals for behavioral health evaluation and the actual number of evaluations would increase to reflect the increase in need. It is counterintuitive that, in fact, the opposite occurred. Despite the increase in behavioral health concerns [8,9], there was a sharp decline in every aspect of service utilization: referrals, students seen, and patient visits. Possible explanations for decreased referral for mental health treatment include lack of teacher/student contact, an absence of teacher awareness of student well-being, and lack of direct walk-in access to the SBHCs. It is also possible that deterioration in mental health in youth, loneliness, and isolation may have decreased the motivation required for help-seeking behavior. Families and the public may have also perceived mental health as less urgent than the imminent risk of a potentially lethal infection. 

Our data suggest that despite a doubling of mental health needs [7,9,28], service utilization in the SBHCs did not recover to pre-pandemic levels at the end of 2021, even after youth returned to school with full access to in-person behavioral health care. When schools reopened to in-person learning in spring 2021, the SBHCs were only offering telehealth and there was no increase in service utilization. Even when school was open, the rates of mental health care in the SBHCs did not increase until the modality of treatment was switched from telehealth to in-person. The modality of treatment (i.e., access to in-person treatment) had a greater impact on service utilization than whether or not the schools themselves were open for in-person learning. 

During the pandemic, while there was a decline in referrals and patient visits, there was also a decrease in terminations and transfers. Two things may have contributed to the decline in terminations. First, it became impossible to transfer patients, even complex patients, because of long waitlists outside of the SBHCs. Second, duration of treatment as reflected by a decrease in terminations may have increased to reflect the increased acuity of youth during the crisis. 

In the fall of 2021, there was an increase in the number of referrals vs. the number of evaluations, suggesting improvement in the SBHCs’ ability to connect to youth needing treatment. This increase in referrals led to longer wait times before starting treatment. Further research is needed to evaluate service utilization through 2022 and 2023 to determine if this trend continues. 

Various other circumstances may have impacted service utilization. Although there has been a decrease in overall school enrollment [25], the numbers of the ‘pandemic missing’ are too small to have had a significant impact on the results reported here. Immediately after returning to school, there may have been a ‘honeymoon’ period, in which the relief of returning to school obscured students’ mental health symptoms. The initial return to school was associated with myriad other concerns such as masking, COVID-19 testing, a shortage of teachers, and other issues which may have overshadowed the need for recognizing and referring youth with behavioral health issues. 

Taken together, our results suggest that telehealth compromised many of the advantages that are uniquely associated with an SBHC: convenience, easy access, and face-to-face support in a familiar setting. Referrals declined, and even students who were distressed did not reach out and self-refer. Adolescents may fail to fully appreciate that their distress can be treated and may be reluctant to seek care on their own without adult or teacher support. The demographic of the student population in this study includes a significant number of children from non-English speaking, immigrant families where mental health stigma is prominent, thus limiting family support for referral. An initial telehealth appointment requires being proactive and planning. This raises the possibility that there might be a preferential advantage for in-person initial evaluations, while students who are already engaged in treatment may benefit from some of the flexibility provided by telehealth. Further research is needed to determine for whom, when, and how to optimize combinations of service modalities. 

Youth who are ‘zoomed out’ with online learning may be resistant to virtual therapy. This is especially true for those students who are vulnerable because they failed to engage with online learning at all. The pandemic was associated with loneliness and mourning the loss of real-life, in person contact. Youth who were isolated in a virtual cocoon during lockdown, who perceive screens as the agent of their loneliness, may resist virtual therapy as the antithesis of the connectedness they are seeking. 

## 5. Limitations

This study has several limitations. The study is localized to three SBHCs within one health care system, and although our data are consistent with national statistics, it is unclear the extent to which our results can be generalized to other SBHCs. Our analyses are limited to aggregate data, which precluded analyses of the impact of moderators such as race, ethnicity, gender, and social determinants of health or insurance, although the demographics of our sample are diverse and include a substantial number of disadvantaged students on Medicaid. Lastly, the data in this study are time-limited to service utilization through the end of December 2021. Ongoing research is needed to evaluate the success of telehealth and its role in service utilization up to the present. Not only does continued research post-pandemic clarify the relative impact of lockdown vs. modality of treatment but it also helps us to determine the extent to which the SBHC are recovering or exceeding pre-pandemic levels of care and approaching the level of care that are required to respond to the youth mental health crisis. 

## 6. Conclusions

The strength of this study is that we were able to look at both the pandemic as well as the impact of modality of treatment on service utilization in SBHCs. If our finding that telehealth compromises some of the unique advantages to delivering care within a SBHC, then further research is needed to identify the role for telehealth within SBHCs. 

Our findings align with other recent publications looking at the impact of the pandemic on youth [27,29] and have important implications for considering future strategies for addressing the global and national youth mental health crisis. When schools and SBHCs are closed, youth are at a greater risk for negative outcomes across domains including physical health (e.g., loss of access to meals and less informal assessment of children outside of the home), social development (e.g., limited opportunities to interact with peers, which is a critical aspect to child development), and academic development (e.g., loss of learning, decreased engagement, etc.). Compounding an increase in risk with a decrease in access to care leads to a vicious cycle that is central to understanding the current youth mental health crisis. In summary, during the pandemic when schools closed, the need for mental health care increased dramatically while service utilization declined. Mental health screening and school-based mental health care needs to increase to remediate the ‘echo pandemic’ of profound and unprecedented levels of distress among youth that followed COVID-19. 

Qualitative studies would be helpful to better understand why youth who are distressed may fail to reach out for help, and to understand how youth feel about in-person vs. telehealth care, particularly in SBHCs. Creative options for hybrid care, and peer-to-peer support may be needed to respond to a social crisis characterized by isolation. This issue is as important now as it was during the pandemic. Research on how to identify and access youth in need and combine modalities of treatment to optimize treatment is essential to remediating and responding to the youth mental health crisis. 

## Figures and Tables

**Figure 1 ijerph-20-04588-f001:**
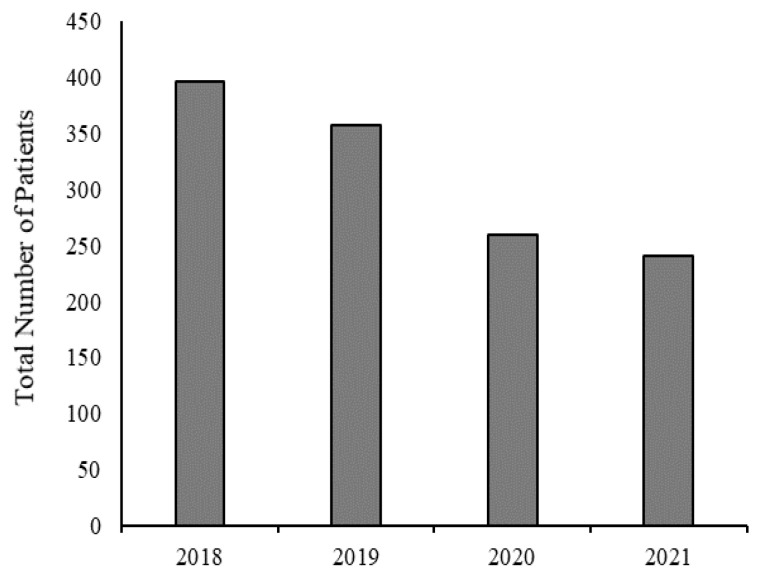
Total number of patients seen for behavioral health services in the SBHCs by year.

**Figure 2 ijerph-20-04588-f002:**
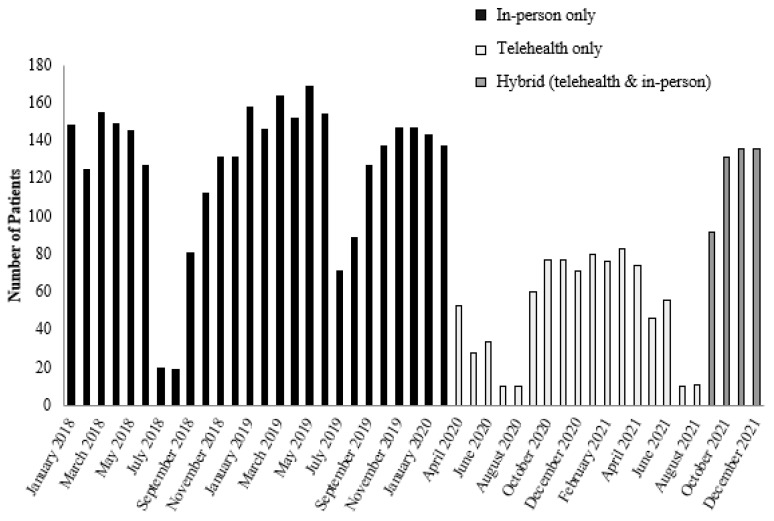
Total number of patients seen by month during periods of in-person care, telehealth only, and hybrid services.

**Figure 3 ijerph-20-04588-f003:**
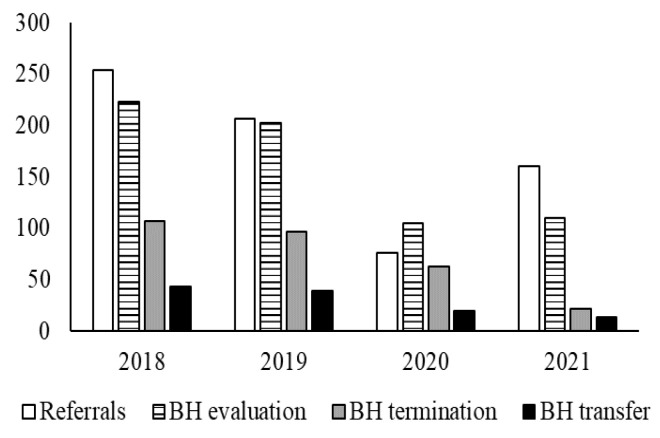
The number of behavioral health evaluations, referrals, terminations, and transfers at the SBHCs from 2018 to 2021.

**Table 1 ijerph-20-04588-t001:** Racial, ethnic, and linguistic demographic information for the students at the three public high schools with SBHCs.

Race, Ethnicity, Language	Cambridge (*n* = 1833)	Everett (*n* = 2081)	Somerville (*n* = 1260)
African American	27.4%	18%	10.9%
Asian	9.8%	5.2%	6.2%
Hispanic	15.2%	55.2%	45.9%
Native American	0.2%	0.5%	0.1%
White	38.5%	19.1%	34.7%
Native Hawaiian, Pacific Islander	0.4%	0.1%	0.2%
Multi-Race, Non-Hispanic	8.4%	1.8%	2.1%
First Language was not English	24.5%	62.6%	54.6%

## Data Availability

Original data are available upon request from the corresponding author.

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
