# Peer review of "The Impact of School Closures on Service Utilization in School-Based Health Centers"

_ijerph, 2023, doi:10.3390/ijerph20054588_

Round 1
Reviewer 1 Report
This manuscript presents descriptive data from several school based health centers on numbers of mental health services provided pre, during, and post COVID-19 pandemic related school closures. The topic is very important, given the child/adolescent mental health crisis. Strengths of the manuscript include a relatively unique dataset, strong results, and good conceptual presentation (review of the problem, current knowledge, and discussion sections). Relative weaknesses include lack of inferential statistics and a limitations section, and need for further clarity in the results section and description of the results in the discussion section. The whole manuscript must also undergo significant editing for numerous grammar and punctuation errors.
More specifically:
Line 101 – provide details for the statement “The percentage offering some telehealth services doubled.”
It would be helpful to have more information about the student populations in the schools included in the study, such as total student body, race/ethnicity of students, etc., even if that information is not available for the patients seen.
The authors should consider using basic inferential statistics to examine whether decreases in number of patients seen, etc., are significant. Several statements in the results section are presented as if they are the results of significance testing, e.g., lines 196-197, “After the schools and SBHCs re-opened, this did not improve significantly, with 2601 visits occurring in 2021.”
Provide details for the statement on lines 156-157 about a 37% decrease – from what year to what year? Are these referring to averages across 2018-2019 and 2020-2021?
The shading in Figures 2 and 3 is difficult to read, especially when printed. Change to increase contrast.
Add headings in the Results section, e.g., 3.2 Patients Seen by Month, 3.3 Behavioral Health Services by Year, 3.4 Number of Patient Visits (before line 194), or alternatively, delete 3.1.
Move the Figure 3 to just after the title at line 183.
Begin the first paragraph of the Discussion section with a restatement of the results.
Several statements in the paragraphs from lines 222 to 250 refer to data not presented in this study. Without more information about that data, it is difficult for the reader to judge whether these statements are adequately supported. For example, lines 225-226 – “Students themselves seem to show less interest in telehealth care than in person care.” What is the evidence for that?
As with the Results section, several statements in the Discussion suggest that significance testing (inferential statistics) was conducted, when it was not, for example, lines 242-244, “…those students who were engaged n care, stayed in treatment longer than they otherwise would have pre-pandemic…”
A limitations section is needed that documents several limitations, including lack of significance testing (if that continues), lack of examination of other factors that may have affected the results, relatively small sample size (e.g., 3 schools), etc.
Author Response
|
Provide details for the statement “the percentage offering some telehealth services doubled.” |
Statement was adjusted to the following: Nevertheless, over half of the SBHCs (57%) initiated or increased telehealth services. 22 |
p.2 |
|
Information about the schools student populations |
Table 1 was added to provide the total number of students at each high school as well as racial, ethnic, and linguistic demographic information about the students. |
p.4 |
|
As with the Results section, several statements in the Discussion suggest that significance testing (inferential statistics) was conducted, when it was not, for example, lines 242-244, “…those students who were engaged n care, stayed in treatment longer than they otherwise would have pre-pandemic…” |
This has been corrected such that inferential statistics were utilized and results were added. |
p.6 |
|
A limitations section is needed that documents several limitations, including lack of significance testing (if that continues), lack of examination of other factors that may have affected the results, relatively small sample size (e.g., 3 schools), etc. |
A limitations section was added (section 5) that outlines the limitations you suggested. |
p.9 |
|
Provide details for the statement on lines 156-157 about a 37% decrease – from what year to what year? Are these referring to averages across 2018-2019 and 2020-2021? |
Additional details were provided to increase the clarity of the statement: There was a 34% decline in the average number of patients seen between 2018-2019 and 2020-2021. |
p. 5 |
|
Shading in Figures 2 and 3 |
The graphs have been revised to be more conducive to black/white coloring & create a uniform presentation. |
p.5, 6, 7 |
|
Add headings in the Results section, e.g., 3.2 Patients Seen by Month, 3.3 Behavioral Health Services by Year, 3.4 Number of Patient Visits (before line 194), or alternatively, delete 3.1. |
Thank you for the suggestion. Headings were added for consistency and organization throughout the Results section. |
p. 5-7 |
|
Move the Figure 3 to just after the title at line 183. |
Figure 3 was moved to just after the title line. |
p.7 |
|
Begin the first paragraph of the Discussion section with a restatement of the results |
The discussion section begins with a restatement of the results– Our findings are relevant to understanding the impact of the pandemic on youth mental health, the demand for service, and the relative effectiveness of in-person, telehealth and hybrid modalities of treatment in SBHCs. Our data indicate that closure of in-person services had a deleterious impact on access to care despite the switch to telehealth. |
p.7 |
|
Several statements in the paragraphs from lines 222 to 250 refer to data not presented in this study. Without more information about that data, it is difficult for the reader to judge whether these statements are adequately supported. For example, lines 225-226 – “Students themselves seem to show less interest in telehealth care than in person care.” What is the evidence for that? |
We have removed these findings from the manuscript, as they were indeed confusing and not backed by data within the current study. |
N/A |
|
As with the Results section, several statements in the Discussion suggest that significance testing (inferential statistics) was conducted, when it was not, for example, lines 242-244, “…those students who were engaged n care, stayed in treatment longer than they otherwise would have pre-pandemic…” |
This has been corrected and inferential analyses were added.
The statement about treatment length has been removed from the manuscript, as this was not backed by data within the current study. |
p. 6 |
Reviewer 2 Report
The purpose and the context of the study are interesting and highly relevant in the post-COVID period. The results show that there is a need to reestablish mental health services in a way that is accessible for young people.
The research data were taken from the health care system. The data presentation is adequate, however, the graphs need to be edited, the visual quality is not sufficient.
In the discussion, the authors mention qualitative interviews and intervention initiatives, however, these are neither properly described and analysed in the Method and Results sections, nor cited as an additional source. This part of the text needs improvement.
Author Response
|
The data presentation is adequate, however, the graphs need to be edited, the visual quality is not sufficient. |
The graphs have been revised to be more conducive to black/white coloring & create a uniform presentation. |
p. 5, 6, 7 |
|
In the discussion, the authors mention qualitative interviews and intervention initiatives, however, these are neither properly described and analyzed in the Method and Results sections, nor cited as an additional source. This part of the text needs improvement. |
The references to qualitative interviews & intervention initiatives refer to a study that was taking place concurrently and have been removed to minimize confusion. |
N/A |
Reviewer 3 Report
The manuscript covers the important theme of the extent to which remote interventions' increasing popularity can cater to the needs of those who need these services. However, the current draft requires significant edits before being considered for publication.
Abstract
- The background doesn’t connect well to the methods.
- Add data to support authors’ claims about increased mental health needs and decreased referrals, evaluations, and services offered.
Introduction
- Authors must read the introduction section closely and focus on streamlining text by avoiding repetition and improving connections among paragraphs.
- In the first para, the authors have included isolation and family issues among both social and emotional stressors. The text can be streamlined. Authors may also want to consider factors like loss of peer connections, and disruption in daily schedule to add richness to the information conveyed.
- The last line of the first para on social angst and compliance among adolescents is difficult to follow. Also, is there literature on the basis of which authors are making this claim?
- In para three, please specify the period the authors refer to as the last two years.
- The authors concluded the last para by stating that bio-psycho-social factors like infection, in-person education, peer connection, etc., must be addressed to successfully resolve the syndemic. Then the fourth and fifth para goes again into the role of schools. Authors need to streamline and present information more comprehensively.
- The way the first para is written gives the impression that WHO called for a targeted school-based program during the pandemic. But work by Hamoda et al was focused on school re-opening rather than the peak of the pandemic. Please review and revise the para accordingly.
- Para 6,7 and 8 on the importance of SBHC can be combined to focus on the core message. Para 9 looks outs of the place.
- In the last para of the introduction, please specify what constitutes the first two years of the pandemic compared with the two years prior to the pandemic. Also, specify what authors mean case-load? Is it active cases per counsellor or per centre? Also, this para starts with a research question about caseload, but the hypothesis is about case presentation. These are different terms. Authors need to be more consitent in use of terminology
Methods
- From the methods, it’s unclear whether this study is a retrospective review of previously collected data or data was collected for the purpose of the study. If data was collected for the purpose of the study, the rationale for the waiver of IRB review is not clear.
- Provide information about the exact period of school closure and re-opening, so it’s easier to understand the following results.
Results
- Clubbing of 2021 telehealth and hybrid service data and making claims about the return to school and its impact on service utilisation seems misguided and confusing.
- În figure 3 description, the authors state that ‘students who were engaged in live treatment before the pandemic were less likely to be terminated’. Its not clear what the authors are trying to assert. Are they implying those who started treatment before the pandemic never reached their successful termination? How is that possible? or are they referring to premature termination? What were the criteria for termination and referral here?
- Authors claim that those students who were engaged in care stayed in treatment longer than they otherwise would have pre-pandemic. Based on three figures, it’s unclear how this conclusion was reached.
- Results don’t cover the ‘change in presentation’ that was given as one of the variables in the introduction.
Discussion
I haven’t reviewed the discussion section, as I think the results require significant reorganisation and review. Overall, I think this may be more suitable for a brief report format rather than a full article.
